# Primary Prevention of Cardiovascular Risk in Lithuania—Results from EUROASPIRE V Survey

**DOI:** 10.3390/medicina56030134

**Published:** 2020-03-18

**Authors:** Gediminas Urbonas, Lina Vencevičienė, Leonas Valius, Ieva Krivickienė, Linas Petrauskas, Gintarė Lazarenkienė, Justina Karpavičienė, Gabrielė Briedė, Emilė Žučenkienė, Karolis Vencevičius

**Affiliations:** 1Department of Family Medicine, Medical Academy, Lithuanian University of Health Sciences, 50161 Kaunas, Lithuania; leonas.valius@lsmuni.lt (L.V.); ieva92@gmail.com (I.K.); linas.petrausk@gmail.com (L.P.); gintare.lazarenkiene@gmail.com (G.L.); justina.griskaite@gmail.com (J.K.); 2Institute of Clinical Medicine, Clinic of Internal Diseases, Family Medicine and Oncology, Faculty of Medicine, Vilnius University, 01513 Vilnius, Lithuania; lina.venceviciene@santa.lt (L.V.); ceponyte.gabriele@gmail.com (G.B.); emile.zucenkiene@gmail.com (E.Ž.); karolis.vencevicius@gmail.com (K.V.)

**Keywords:** EUROASPIRE V, cardiovascular risk, primary prevention, cardiovascular risk perception

## Abstract

*Background and Objectives:* Cardiovascular disease (CVD) prevention guidelines define targets for lifestyle and risk factors for patients at high risk of developing CVD. We assessed the control of these factors, as well as CVD risk perception in patients enrolled into the primary care arm of the European Action on Secondary and Primary Prevention by Intervention to Reduce Events (EUROASPIRE V) survey in Lithuania. *Materials and Methods:* Data were collected as the part of the EUROASPIRE V survey, a multicenter, prospective, cross-sectional observational study. Adults without a documented CVD who had been prescribed antihypertensive medicines and/or lipid-lowering medicines and/or treatment for diabetes (diet and/oral antidiabetic medicines and/or insulin) were eligible for the survey. Data were collected through the review of medical records, patients’ interview, physical examination and laboratory tests. *Results:* A total of 201 patients were enrolled. Very few patients reached targets for low-density lipoprotein cholesterol (LDL-C) (4.5%), waist circumference (17.4%) and body mass index (15.4%). Only 31% of very high CVD risk patients and 52% of high-risk patients used statins. Blood pressure target was achieved by 115 (57.2%) patients. Only 21.7% of patients at very high actual CVD risk and 27% patients at high risk correctly estimated their risk. Of patients at moderate actual CVD risk, 37.5% patients accurately self-assessed the risk. About 60%–80% of patients reported efforts to reduce the intake of sugar, salt or alcohol; more than 70% of patients were current nonsmokers. Only a third of patients reported weight reduction efforts (33.3%) or regular physical activity (27.4%). *Conclusions:* The control of cardiovascular risk factors in a selected group of primary prevention patients was unsatisfactory, especially in terms of LDL-C level and body weight parameters. Many patients did not accurately perceive their own risk of developing CVD.

## 1. Introduction

With over 17 million deaths globally each year, cardiovascular diseases (CVDs) are the leading cause of morbidity and mortality worldwide [1,2]. In Europe, over four million deaths due to CVD occur each year, accounting for 45% of the total mortality [3]. Ischemic heart disease (IHD), the most prevalent CVD, accounted for approximately 1.8 million deaths [3]. From 2003 onwards, a decrease in age-standardized death rates for CVD and IHD were observed in most of the European countries [3]. In Lithuania, mortality due to IHD decreased by 14% in men and by 18% in women over the 10-year period [3]. However, an inequality in CVD mortality declines exists in Europe, with the largest differences between lower and higher socioeconomic groups observed in Central-Eastern European and Baltic countries [4].

CVD can be prevented through lifestyle and pharmacological interventions both at the population and individual level [5]. Established modifiable risk factors for CVD include hypertension, smoking, elevated lipid levels, diabetes, diet, obesity, sedentary behavior and physical activity [6]. Reduction of at least some of these factors favorably impacts CVD risk. Each 1-mmol/L (38.7 mg/dL) reduction in low-density lipoprotein cholesterol (LDL-C) significantly reduces the risk of major vascular events by ~20% [7], even in patients with baseline LDL-C levels as low as a median 1.6 mmol/L (63 mg/dL) [8]. Long-term CVD mortality risk increases by half for every 20-mm Hg increase in systolic blood pressure above 115 mm Hg [9]. Reduction of blood pressure protects against major cardiovascular events in both younger (<65 years) and older (≥65 years) individuals [10]. Changes in three major cardiovascular risk factors (smoking, serum cholesterol and systolic blood pressure) were estimated to contribute to about two-thirds of the reduction in IHD mortality [11]. The presence of type 2 diabetes increases the risk of IHD, ischemic stroke and vascular deaths by approximately two-fold [12]. Although the effect of diabetes control on the reduction of cardiovascular outcomes is not as noticeable as in the case of other modifiable risk factors [13], tight glycemic control is recommended [14]. Metabolic syndrome, a complex disorder of several interconnected factors such as abdominal obesity, atherogenic dyslipidemia (elevated triglycerides and reduced high-density lipoprotein cholesterol (HDL-C)), elevated blood pressure, insulin resistance (with or without glucose intolerance), proinflammatory and prothrombotic states are associated with a two-fold increase in cardiovascular outcomes and a 1.5-fold increase in all-cause mortality [15,16]. Lifestyle modification with an emphasis on weight reduction is the main goal in the management of metabolic syndrome [17]. Even weight loss in a range of 10% can reduce CVD risk [18,19]. However, metabolic syndrome does not set thresholds either for LDL-C or for total cholesterol [15].

Since the mid-1990s, the Joint European Societies’ (JES) guidelines on CVD prevention have been periodically revised to define targets for lifestyle and risk factor controls for both patients with IHD and those at high risk of developing CVD. The implementation of the JES guidelines in clinical practice has been assessed in five cross-sectional EUROASPIRE (European Action on Secondary and Primary Prevention by Intervention to Reduce Events) surveys. The number of participating countries increased from nine countries in the first EUROASPIRE survey to 27 countries in the fifth survey. Starting from the EUROASPIRE III, the surveys also included a primary care arm, i.e., individuals at high CVD risk because of hypertension, dyslipidemia or type 2 diabetes mellitus (T2DM) [20].

In this article, we present data on the control rates of lifestyle and risk factors, as well as CVD risk perception in Lithuanian patients enrolled in the primary care arm of the fifth EUROASPIRE (EUROASPIRE V) survey.

## 2. Materials and Methods

Data were collected as a part of the EUROASPIRE V survey, a multicenter, prospective, cross-sectional observational study carried out in 2016–2017 in 27 European countries. In Lithuania, a primary care arm of the study was conducted at two primary health care centers: the Clinic of Family Medicine at the Hospital of the Lithuanian University of Health Sciences Kauno Klinikos (Kaunas) and the Center of Family Medicine at the Vilnius University Hospital Santaros Klinikos (Vilnius). The study was approved by the Lithuanian Bioethics Committee (issue number, L-16-06/02, 8 September 2016).

The inclusion criteria were adult (18–80 years old) people without a documented CVD who had been prescribed one or more of the following treatments for at least six months and not more than 24 months: antihypertensive medicines and/or lipid-lowering medicines and/or treatment for diabetes (diet and/or oral antidiabetic medicines and/or insulin). Consecutive patients were identified through the review of medical records at participating health care centers and were invited personally to participate in the study. The enrollment was continued until the target of 100 patients was reached at each center. All patients provided written informed consent.

For each patient, specially trained investigators recorded the following data using standardized questionnaires: sociodemographic data, lifestyle risk factors, medical history, treatment and CVD risk perception. The following objective measurements were performed: height, weight and waist circumference, blood pressure and heart rate. Blood samples were taken for hematology and clinical chemistry. Blood samples were analyzed at local laboratories.

The total actual CVD risk was assessed using criteria proposed by the European Society of Cardiology (ESC) and European Atherosclerosis Society (EAS) [21]:-Very high CVD risk: patients with any of the following: a) diabetes with target organ damage or at least three major risk factors or early onset of type 1 diabetes of a duration of >20 years, b) severe chronic kidney disease (estimated glomerular filtration rate of <30 mL/min/1.73 m^2^), c) a calculated Systematic Coronary Risk Estimation (SCORE) ≥10% for a 10-year risk of fatal CVD or d) familial hypercholesterolemia with atherosclerotic CVD or with another major risk factor.-High CVD risk: patients with markedly elevated single risk factors, familial hypercholesterolemia without other major risk factors, diabetes without target organ damage, with a diabetes duration ≥10 years or another additional risk factor, moderate chronic kidney disease (estimated glomerular filtration rate of 30–59 mL/min/1.73 m^2^) and a calculated SCORE ≥5% and <10%.-Moderate CVD risk: young patients (type 1 diabetes <35 years or type 2 diabetes <50 years) with a diabetes duration <10 years without other risk factors and a calculated SCORE ≥1% and <5%.-Low CVD risk: a calculated SCORE <1%.

For the diagnosis of metabolic syndrome, at least three of the following criteria had to be met [15]: fasting plasma glucose ≥5.6 mmol/L, waist circumference >102 cm in men and >88 cm in women, triglycerides ≥1.7 mmol/L, HDL-C <1.0 mmol/L in men and <1.3 mmol/L in women or pharmacological treatment for low HDL-C and blood pressure ≥130/85 mmHg or pharmacological treatment for hypertension.

To assess the CVD risk perception, patients were asked the following five questions:-Do you think everyone needs to know their risk of having heart disease, stroke or another vascular disease to reduce the risk? The possible responses were “strongly disagree”, “disagree”, “neither agree nor disagree”, “agree” or “strongly agree”.-Are you worried that you may have heart disease, stroke or another vascular disease? The possible responses were “strongly disagree”, “disagree”, “neither agree nor disagree”, “agree” or “strongly agree”.-Are you afraid to find out what is your risk of having heart disease, stroke or another vascular disease? The possible responses were “strongly disagree”, “disagree”, “neither agree nor disagree”, “agree” or “strongly agree”.-What is your risk of developing heart disease, stroke or another vascular disease in the next 10 years? The possible responses were “very low”, “low”, “moderate”, “high”, “very high” or “do not know”.-Do you think your risk of developing heart disease, stroke or another vascular disease in the next 10 years is higher, lower or the same as that of other people of your gender and age? The possible responses were “much higher”, “higher”, “same”, “lower” or “much lower”.

To assess efforts to reduce CVD risk, patients were asked the following question: “Over the past three years, what actions have you taken to reduce your risk of cardiovascular diseases?” The list of possible answers included reduced salt intake, reduced calorie intake, reduced sugar intake and reduced alcohol intake. To assess physical activity, patients were asked the following question: “Do you take a regular physical activity of 20–60-min duration on average three to five times a week?” To assess weight reduction, patients were asked the following question: “Over the past month, have you actively tried losing your body weight?”

The main outcome measures were the proportions of the patients achieving lifestyle, risk factor and therapeutic targets. The following CVD prevention targets were used:-LDL-C: <1.4 mmol/L for individuals at very high CVD risk, <1.8 mmol/L for individuals at high risk, <2.6 mmol/L for individuals at moderate risk and <3.0 mmol/L for individuals at low risk (targets recommended by the 2019 ESC/EAS dyslipidemia management guidelines [21]);-triglycerides: <1.7 mmol/L (a target recommended by 2019 ESC/EAS dyslipidemia management guidelines [21]);-body weight: body mass index (BMI), 20–25 kg/m^2^, waist circumference <94 cm (men) and <80 cm (women) (targets recommended by 2016 JES guidelines on CVD prevention [6]);-blood pressure: <140/90 mmHg (a target recommended by 2016 JES guidelines on CVD prevention [6]) and <130/80 mmHg (a target recommended by 2018 ESC/European Society of Hypertension (ESH) hypertension guidelines [22]).

The descriptive statistics were applied for the data analysis. Normality of distribution was tested using a Kolmogorov-Smirnov test (for a sample size >50) and Shapiro-Wilk tests (for a sample size between 30 and 50). Categorical data were presented as a count (*n*) and percentage (%). Continuous variables were presented as median (first quartile (Q1) and third quartile (Q3)). Chi-square test was used to compare categorical variables. Nonparametric Mann-Whitney U test was used to compare continuous variables. Statistical differences were interpreted at a 5% (two-sided) significance level. Data were analyzed using the statistical software package SPSS version 20.

## 3. Results

In Lithuania, 201 patients (73 men and 128 women, median age 59.0 years) were enrolled into the primary care arm of the EUROASPIRE V survey. Of them, 105 (52.2%) patients were enrolled because of treatment with antihypertensive medicines, 53 (26.4%) patients were enrolled because of lipid-lowering medicines and 43 (21.4%) patients were enrolled because of treatment for diabetes (patients might have been using one or more of the treatments listed above). Most of patients were highly educated. However, almost 30% of patients were unemployed, and 24% of patients reported low or very low incomes. All baseline sociodemographic and clinical characteristics are summarized in Table 1.

About 60% of patients were at a very high or high CVD risk. Among the patients at very high risk, there were 34 (73.9%) patients with diabetes and 15 (32.6%) patients with a calculated SCORE ≥10%. Metabolic syndrome was present in nearly 65% of patients (Table 1).

Proportions of the patients who achieved the recommended cardiovascular risk factor targets are reported in Table 2. Very few patients reached targets for LDL-C and body weight parameters (waist circumference and BMI). Only 31% of very high CVD risk patients and 52% of high-risk patients used statins. The use of statins was significantly more common among patients with metabolic syndrome compared to those without metabolic syndrome. The median (Q1 and Q3) LDL-C concentration was significantly lower in patients taking statins than in those who did not use statins—2.9 (2.3, 3.6) mmol/L vs. 4.0 (3.4, 4.6) mmol/L, respectively (*p* < 0.001).

Suboptimal blood pressure control was observed in more than half of patients, irrespective of their CVD risk or the presence of the metabolic syndrome. Even fewer patients were able to achieve a new blood pressure target of <130/80 mmHg recently recommended by the ESC/ESH hypertension guidelines [22].

Most of the patients (89.6%) agreed that CVD risk awareness is important; 76.1% of patients were concerned that it might affect them personally (Table 3). However, 34.3% of patients were afraid of knowing their own risk. When asked about their CVD risks, only 8.0% and 25.9% of patients attributed themselves to a very high or high-risk group, respectively. Less than half of the patients (46.3%) assumed that their CVD risk is much higher or higher than that of other people of their age and gender.

The proportion of the patients underestimating their own CVD risk was high; about 50%–60% of the patients at high or very high actual CVD risk reported being at very low, low or moderate risk (Table 4). Only 21.7% of the very high-risk patients and 27% of the high-risk patients correctly estimated their risk. Of the patients at moderate actual CVD risk, 37.5% patients accurately self-assessed the risk. Patients estimated their own CVD risk similarly, irrespective of the presence of metabolic syndrome.

When asked about the actions taken to reduce the CVD risk, about 60%–80% of patients reported efforts to reduce the intake of sugar, salt or alcohol (Table 5). More than 70% of patients were currently nonsmoking. Only a third of patients reported weight reduction efforts or regular physical activity. There were no clear trends in the self-reported efforts within CVD risk groups or between patients with or without metabolic syndrome, except for the lower prevalence of nonsmokers among very high CVD risk patients and lower weight reduction efforts among patients with metabolic syndrome.

## 4. Discussion

In patients enrolled into the primary care arm of the EUROASPIRE V survey in Lithuania, we found an unsatisfactory cardiovascular risk factor control. Especially low control rates were found for LDL-C (4.5%) and body weight parameters (15%). Numerous observational studies, including previous EUROASPIRE surveys, concluded that control of multiple cardiovascular risk factors is poor in clinical practice [20,23,24,25,26,27], and it is even more difficult to achieve the targets in the primary prevention setting than in coronary patients [20]. In the primary care arm of EUROASPIRE IV, less than half (43%) of the treated patients reached the blood pressure target, one-third (33%) of patients achieved the LDL-C target of <2.5 mmol/L and less than two-thirds (59%) achieved the HbA1c target of <7.0%. No major differences in the control of lifestyle and risk factors in people at high risk of developing CVD were observed between the EUROASPIRE III and IV surveys [20]. The European Study on Cardiovascular Risk Prevention and Management in Usual Daily Practice (EURIKA) reported that among outpatients with varying degrees of CVD risk, less than half of the treated hypertensive and dyslipidemic patients and a third of the diabetic patients attained treatment goals. Only a quarter of patients achieved a satisfactory BMI and waist circumference [24]. A retrospective Italian study found a poor control of major cardiovascular risk factors (diastolic blood pressure, LDL-C, HDL-C, triglycerides and BMI) in outpatients with high or very high SCORE risks, despite the greater use of pharmacological treatment and lifestyle recommendations [26]. In a cohort of male workers with no previous CVD, only 46% of the workers with hypertension, 22% of those with diabetes and 11% of those with dyslipidemia were controlled [23]. In a large outpatient cohort in Italy, LDL-C levels were poorly controlled despite lipid-lowering therapy and lifestyle recommendations; none of the primary prevention patients with very high SCORE risks achieved the LDL-C target of <70 mg/dL (<1.8 mmol/L), and only 9% of high SCORE risk patients had an LDL-C of <100 mg/dL (2.6 mmol/L) [27]. In countries outside Western Europe, the proportion of high or very high CVD risk patients achieving treatment targets was 44% for LDL-C, 56% for blood pressure and 39% for diabetes, but only one-tenth of patients achieved simultaneous control of LDL-C, diabetes and blood pressure [25].

In our study, the proportion of patients achieving the blood pressure target of <140/90 mmHg was similar to that reported in other studies, whereas the control of LDL-C was much worse. The possible reason might be the stricter LDL-C target levels chosen for our analysis; we used recently proposed goals for intensive LDL-C reduction across CVD risk categories [21], whereas earlier studies relied on higher LDL-C target levels. Besides, we collected data at the time when physicians followed the 2016 ESC/EAS lipid guidelines and aimed to achieve less stringent treatment targets (i.e., <1.8 mmol/L for individuals at very high CVD risk, <2.6 mmol/L for individuals at high risk and <3.0 mmol/L for individuals at low to moderate risk [28]). Another explanation of poor lipid control is low statin use. We found that only 30% of the very high CVD risk patients and 50% of the high CVD risk patients took statins, while the use of antihypertensive agents was reported by 90% of patients. Of note, the use of statins was far more prevalent (76%) among patients with metabolic syndrome, suggesting that in Lithuania, metabolic syndrome is still a more important factor than the total CVD risk assessed using the ESC/EAS criteria. The possible reason might be the national financial program for the early detection of CVD, which recommends referring patients with metabolic syndrome to specialized CVD prevention centers [29].

Although controlled clinical studies proved the efficacy of primary prevention with statins in reducing all-cause mortality and the risk of primary cardiovascular events [30,31], other observational studies also reported the undertreatment of patients eligible for preventive lipid-lowering therapy [27,32,33,34,35,36]. The rates of undertreatment with antihypertensive agents are lower [36,37,38], which can be explained by well-established evidence on efficacy, good safety profiles and the low cost of these drugs [37]. Meanwhile, some controversy regarding the risks and benefits of statins used in primary prevention still exists both in scientific literature [39] and clinical practice. The causes of statin underuse involve both clinician- and patient-related factors. Barriers reported by clinicians include concerns about cost-effectiveness, safety, the medicalization of healthy people, patient adherence, negative effects on health behavior, variable lipid targets, complex CVD risk assessment tools and impact on workload [40]. Lack of adherence and persistence to treatment are the main patient-related barriers [41]. Adherence to statins is suboptimal, both among patients with established CVD and those at high CVD risk [42,43]. The rates of statin discontinuation range from 41% in secondary prevention to 47% in primary prevention [44]. In the primary prevention setting, the reported proportion of patients defined as adherent to the prescribed statin regimen ranges from 18%–79% [45]. The adherence to cardiovascular preventive medications in general range from 50% in primary prevention studies to 66% in secondary prevention studies, and it is little related to the drug class (aspirin, blood pressure-lowering drugs or statins), suggesting that safety is not the main cause [46]. Traditional cardiovascular risk factors (i.e., older age, male gender, diabetes and hypertension); higher incomes; education level and employment status were positively associated with statin adherence, whereas alcohol abuse and obesity were associated with lower adherence [45]. Patient-reported concerns and barriers to statin adherence include safety concerns; doubts about health benefits; lifestyle preferences; signifying illnesses; medical distrust and logistical barriers (i.e., forgetfulness, costs and regimen) [42,43,47].

We observed no clear trends in the self-reported lifestyle behaviors within CVD risk groups or between patients with or without metabolic syndrome. In addition to other reasons, these findings might be related to the unawareness of their own CVD risk.

Awareness and the accurate perception of their own risk to develop CVD are important to ensure patient’s adherence to pharmacological interventions and lifestyle modification recommendations [48,49,50] and, correspondingly, the achievement of better clinical outcomes. The patients in our study did not accurately perceive their risks of developing CVD. Patients at high or very high actual CVD risk tended to underestimate their risk. Our findings are in accordance with previous researches in various conditions, which reported that patients do not correctly perceive their own risk irrespective of their actual risk [48]. Among patients at high risk for developing CVD, 57% underestimated their risk, and a very similar proportion of low/moderate-risk patients (56%) overestimated the risk [48]. In a Brazilian study, asymptomatic subjects with higher CVD risk according to the Framingham risk score or the lifetime risk score were more likely to underestimate their risk than subjects at intermediate risk [51]. Sociodemographic (i.e., age, gender and socioeconomic status); clinical (i.e., presence and/or treatment for hypertension, diabetes or dyslipidemia and family history of premature myocardial infarction) and psychological (i.e., depressive symptoms, subjective perception of stress and personal health) factors might be related to an inaccurate CVD risk perception [49,50,52,53,54].

Being the frontline care providers in close relationships with patients, primary health care specialists have a unique opportunity to increase patients’ awareness and perception of health risks. When counselling patients at high or very high CVD risk, family physicians should repeatedly comunicate the elevated risk of cardiovascular events and discuss risk factors with the message that risk reduction is possible. A long-term risk factors management plan should be made individually, and physicians, community nurses, and patients should work as a team focusing on its implementation. CVD prevention targets should be closely monitored and corrective actions taken in a timely manner. Our findings imply that special focus is needed to improve the adherence to and persistence with statin therapy, as well as body weight control measures.

Our study had some limitations. The number of patients investigated was small, and they were enrolled in the health care centers located in the two largest Lithuanian cities. Besides, study participants might be more adherent and willing to be involved in their health and care than patients who did not accept the invitation to join the study. All these factors limit the generalizability of our study findings to all high CVD risk patients in Lithuania. Nevertheless, our results are comparable to those reported by large observational studies. The assessment of risk factors’ control and therapeutic targets was based on objective measurements, and this is the most important strength of this study.

## 5. Conclusions

The control of cardiovascular risk factors in the primary prevention patients who participated in the EUROASPIRE V survey in Lithuania was unsatisfactory, especially in terms of LDL-C level and body weight parameters. Many patients did not accurately perceive their own risk for developing CVD. These findings imply that primary health care specialists should exploit more effectively their opportunities to communicate CVD risks to patients, to set a risk factor management plan and monitor its implementation.

## Figures and Tables

**Table 1 medicina-56-00134-t001:** Sociodemographic and clinical characteristics of the study population.

Characteristics	
Sociodemographic characteristics	
Gender, *n* (%)	
women	128 (63.7)
men	73 (36.3)
Age, median (Q1, Q3 *), years	59.0 (50.0–65.0)
Age groups, *n* (%)	
<45 years	22 (10.9)
45–54 years	46 (22.9)
55–64 years	77 (38.3)
≥65 years	56 (27.9)
Education, *n* (%)	
basic	9 (4.5)
secondary	38 (18.9)
high (university or college)	154 (76.6)
Employment, *n* (%)	
employed	132 (65.7)
partially employed	11 (5.5)
unemployed	58 (28.9)
retired	57 (28.4)
Self-reported income level, *n* (%)	
very low	6 (3.0)
low	43 (21.4)
moderate	149 (74.1)
high	3 (1.5)
Vital signs	
Systolic blood pressure, median (Q1, Q3), mmHg	136.0 (128.0–146.3)
Diastolic blood pressure, median (Q1, Q3), mmHg	88.0 (81.8–94.0)
Heart rate, median (Q1, Q3), beats/min	73.0 (67.0–81.0)
Laboratory parameters	
Total cholesterol, median (Q1, Q3), mmol/L	5.8 (5.0–6.8)
Low-density lipoprotein cholesterol, median (Q1, Q3), mmol/L	3.8 (3.1–4.5)
High-density lipoprotein cholesterol, median (Q1, Q3), mmol/L	1.3 (1.1–1.7)
Triglycerides, median (Q1, Q3), mmol/L	1.4 (0.9–2.0)
Total cardiovascular risk	
Low risk	16 (8.0)
Moderate risk	64 (31.8)
High risk	75 (37.3)
Very high risk	46 (22.9)
Medical history	
Metabolic syndrome, *n* (%)	130 (64.7)
Diabetes, *n* (%)	55 (27.4)
Impaired glucose tolerance **, *n* (%)	18 (9.0)
Impaired fasting glycemia ***, *n* (%)	23 (11.4)
SCORE ^†^, median (Q1, Q3)	2.0 (1.0–4.5)

* First and third quartiles, ** fasting plasma glucose <7 mmol/L and 2 h after glucose load ≥7.8 to <11.1 mmol/L, *** fasting plasma glucose ≥6.1 to <6.9 mmol/L and 2 h after glucose load <7.8 mmol/L and ^†^ systematic coronary risk estimation (SCORE).

**Table 2 medicina-56-00134-t002:** Proportion of the patients achieving lifestyle, risk factor and therapeutic targets.

	All Patients *N* = 201	Cardiovascular Risk Group				Metabolic Syndrome	
		Low Risk *N* = 16	Moderate Risk *N* = 64	High Risk *N* = 75	Very High Risk *N* = 46	Yes *N* = 130	No *N* = 71
Low-density lipoprotein cholesterol target *	9 (4.5)	3 (18.8)	4 (6.2)	1 (1.3)	1 (2.2) ^†^	not applicable	not applicable
Triglycerides <1.7 mmol/L **	126 (62.7)	14 (87.5)	42 (65.6)	45 (60.0)	25 (54.3)	60 (46.2) ^†^	66 (93.0)
Statins use, *n* (%)	42 (20.9)	2 (4.8)	5 (11.9)	22 (52.4)	13 (31.0) ^†^	32 (76.2) ^†^	10 (14.1)
Waist circumference <94 cm (men) and <80 cm (women) **	35 (17.4)	5 (31.2)	13 (20.3)	14 (18.7)	3 (6.5)	5 (3.8) ^†^	30 (42.3)
Body mass index 20–25 kg/m^2^ **	31 (15.4)	7 (43.8)	5 (7.8)	16 (21.3)	3 (6.5) ^†^	7 (5.4) ^†^	24 (33.8)
Blood pressure <140/90 mmHg **	115 (57.2)	8 (50.0)	34 (53.1)	48 (64.0)	25 (54.3)	73 (56.2)	42 (59.2)
Blood pressure <130/80 mmHg ***	39 (19.4)	4 (25.0)	11 (17.2)	15 (20.0)	9 (19.6)	18 (13.8) ^†^	21 (29.6)

Values are numbers (percentages). * <1.4 mmol/L for individuals at very high cardiovascular risk, <1.8 mmol/L for individuals at high risk, <2.6 mmol/L for individuals at moderate risk and <3.0 mmol/L for individuals at low risk (as per the 2019 European Society of Cardiology/European Atherosclerosis Society (ESC/EAS) guidelines for the management of dyslipidemias). ** 2016 Joint European Societies’ (JES) guidelines on cardiovascular disease (CVD) prevention. *** 2018 ESC/ESH guidelines for the management of arterial hypertension. ^†^
*p* < 0.05 for differences among cardiovascular risk groups or between groups according to the presence of metabolic syndrome.

**Table 3 medicina-56-00134-t003:** Cardiovascular risk perception.

Questions on Cardiovascular Risk Perception	All Patients *N* = 201
Do you think everyone needs to know their risk of having heart disease, stroke or another vascular disease to reduce the risk?	
Agree or strongly agree	180 (89.6)
Are you worried that you may have heart disease, stroke or another vascular disease?	
Agree or strongly agree	153 (76.1)
Are you afraid to find out what is your risk of having heart disease, stroke or another vascular disease?	
Agree or strongly agree	69 (34.3)
What is your risk of developing heart disease, stroke or another vascular disease in the next 10 years? *	
Very low or low	16 (8.0)
Moderate	90 (44.8)
High	52 (25.9)
Very high	16 (8.0)
Do not know	26 (12.9)
Do you think your risk of developing heart disease, stroke or another vascular disease in the next 10 years is higher, lower or the same as that of other people of your gender and age? *	
Higher or much higher	93 (46.3)
Same	81 (40.3)
Much lower or lower	26 (12.9)

Data are presented as a number (percentage) of the patients providing a specific answer. * One patient did not answer this question.

**Table 4 medicina-56-00134-t004:** Cardiovascular risk perception.

What Is Your Risk of Developing Heart Disease, Stroke or Another Vascular Disease in the Next 10 Years?	Cardiovascular Risk Group				Metabolic Syndrome	
	Low Risk *N* = 16	Moderate Risk *N* = 64	High Risk *N* = 75	Very High Risk *N* = 46	Yes *N* = 130	No *N* = 71
Very low or low	0	9 (14.1)	3 (4.1)	4 (8.7)	8 (6.2)	8 (11.3)
Moderate	7 (43.8)	24 (37.5)	40 (54.1)	19 (41.3)	60 (46.5)	30 (42.3)
High	7 (43.8)	17 (26.6)	20 (27.0)	8 (17.4)	34 (26.4)	18 (25.4)
Very high	0	2 (3.1)	4 (5.4)	10 (21.7)	14 (10.9)	2 (2.8)
Do not know	2 (12.5)	12 (18.8)	7 (9.5)	5 (10.9)	13 (10.1)	13 (18.3)

Data are presented as a number (percentage) of the patients providing a specific answer.

**Table 5 medicina-56-00134-t005:** Self-reported actions to reduce cardiovascular risk.

Actions to Reduce Cardiovascular Risk	All Patients *N* = 201	Cardiovascular Risk Group				Metabolic Syndrome	
		Low Risk *N* = 16	Moderate Risk *N* = 64	High Risk *N* = 75	Very High Risk *N* = 46	Yes *N* = 130	No *N* = 71
Reduced salt intake	141 (70.1)	13 (81.2)	39 (60.9)	50 (66.7)	39 (84.8)	98 (75.4)	43 (60.6)
Reduced sugar intake	160 (79.6)	11 (68.8)	50 (78.1)	59 (78.7)	40 (87.0)	51 (71.8)	109 (83.8)
Reduced alcohol intake	124 (61.7)	10 (62.5)	42 (65.6)	42 (56.0)	30 (65.2)	39 (54.9)	85 (65.4)
No current smoking	144 (71.6)	10 (62.5)	33 (51.6)	52 (69.3)	17 (37.0) *	75 (57.7)	37 (52.1)
Weight reduction efforts	67 (33.3)	7 (43.8)	21 (32.8)	25 (33.3)	14 (30.4)	77 (59.2) *	57 (80.3)
Regular physical activity	55 (27.4)	6 (37.5)	19 (29.7)	19 (25.3)	11 (23.9)	31 (23.8)	24 (33.8)

Values are numbers (percentages). * *p* < 0.05 for differences among cardiovascular risk groups or between groups according to the presence of metabolic syndrome.

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
