# Peer review of "Primary Prevention of Cardiovascular Risk in Lithuania—Results from EUROASPIRE V Survey"

_medicina, 2020, doi:10.3390/medicina56030134_

Round 1

Reviewer 1 Report

This manuscript entitled "Primary Prevention of Cardiovascular Risk in Lithuania–Results from EUROASPIRE V Survey" by Dr. Urbonas et al assessed the control of targets for lifestyle, risk factors and CVD risk perception in patients enrolled into the primary care arm of EUROASPIRE V survey in Lithuania. Although the control of cardiovascular risk factors in a selected group of primary prevention patients was unsatisfactory, regarding LDL-C level and body weight, this study showed clinical relevance and significance. Overall, this manuscript was well prepared and organized. I have minor concerns:

1) In table 3, for the the number of the last questions is 200, which is not consistent with the N = 201. Can you double check the number of patients in each group and correct it?

2) Are there any suggestions / discussion to improve the patients more correctly perceive their risk to develop CVD and adhere to the treatment?

Author Response

Reviewer 1

This manuscript entitled "Primary Prevention of Cardiovascular Risk in Lithuania–Results from EUROASPIRE V Survey" by Dr. Urbonas et al assessed the control of targets for lifestyle, risk factors and CVD risk perception in patients enrolled into the primary care arm of EUROASPIRE V survey in Lithuania. Although the control of cardiovascular risk factors in a selected group of primary prevention patients was unsatisfactory, regarding LDL-C level and body weight, this study showed clinical relevance and significance. Overall, this manuscript was well prepared and organized. I have minor concerns:

1) In table 3, for the number of the last questions is 200, which is not consistent with the N = 201. Can you double check the number of patients in each group and correct it?

Answer: one of patients did not answer two questions on cardiovascular risk perception. In Table 3, we have added an asterix symbol at the end of these questions with a footnote with the corresponding explanation.

2) Are there any suggestions / discussion to improve the patients more correctly perceive their risk to develop CVD and adhere to the treatment?

 Answer: we have added our suggestions into Discussions section, lines 305-313.

Reviewer 2 Report

Urbonas et al. submitted a well-written and structured, appropriately concise manuscript presenting data from the Lithuanian primary care arm of EURASPIRE V.

The reported results are in accordance with the pertinent literature on this subject matter; the authors also did a good job in addressing and elaborating on regional distinctive features.

With the overall conclusion of CVD risk factors control in the examined primary care patients being unsatisfactory, in this reviewer's opinion the authors should discuss or at least hint to viable corrective strategies.

Finally, please expand the Conclusion (which is way too short).

Author Response

Reviewer 2

Comments and Suggestions for Authors

Urbonas et al. submitted a well-written and structured, appropriately concise manuscript presenting data from the Lithuanian primary care arm of EURASPIRE V.

The reported results are in accordance with the pertinent literature on this subject matter; the authors also did a good job in addressing and elaborating on regional distinctive features.

With the overall conclusion of CVD risk factors control in the examined primary care patients being unsatisfactory, in this reviewer's opinion the authors should discuss or at least hint to viable corrective strategies.

Answer: we have added our suggestions into Discussions section, lines 305-313.

Finally, please expand the Conclusion (which is way too short).

Answer: we have expanded conclusions with a summary of suggested corrective strategy, lines 325-328.